# In Vitro Propagation and Genetic Uniformity Assessment of *Manglietiastrum sinicum*: A Critically Endangered Magnoliaceae Species

**DOI:** 10.3390/plants12132500

**Published:** 2023-06-30

**Authors:** Yiyang Luo, Keyuan Zheng, Xiaodi Liu, Jialu Tao, Xugao Sun, Yanwen Deng, Xiaomei Deng

**Affiliations:** 1Guangdong Key Laboratory for Innovative Development and Utilization of Forest Plant Germplasm, Guangzhou 510642, China; rialaw@163.com (Y.L.); 17854228573@163.com (X.L.); taojialu6413@163.com (J.T.); sunxugao1998@163.com (X.S.); hinmentang@gmail.com (Y.D.); 2College of Forestry and Landscape Architecture, South China Agricultural University, Guangzhou 510642, China; 3Shanghai Key Laboratory of Plant Functional Genomics and Resources, Shanghai Chenshan Botanical Garden, Shanghai 201602, China; kyzheng@cemps.ac.cn

**Keywords:** semi-lignified stems as explants, plant tissue culture, bud induction, bud proliferation, rooting, culture conditions, genetic fidelity

## Abstract

*Manglietiastrum sinicum* Y.W. Law is a critically endangered species with great ornamental and commercial value, which urgently requires protection. We tested different combinations of basal media and plant growth regulators to determine (i) the optimal conditions for bud induction and proliferation of explants and (ii) optimal rooting conditions. RAPD- and ISSR-PCR were used to assess the genetic fidelity of regenerated plantlets. Murashige and Skoog medium (MS) supplemented with 0.5 mg/L 6-benzyladenine (BA) and 0.05 mg/L indole-3-butyric acid (IBA) is the optimal medium for bud induction (100% induction). MSM medium (a special basal medium for *M. sinicum*) was more suitable for the efficient proliferation and rooting of *M. sinicum.* Maximum bud proliferation rate (446.20%) was obtained on MSM, with 0.4 mg/L BA, 0.5 mg/L kinetin, and 0.06 mg/L IBA, while maximum root induction rate (88.89%) was obtained on MSM supplemented with 0.4 mg/L 1-naphthylacetic acid and 1.0 mg/L IBA with a 7-day initial darkness treatment. The rooted plantlets were transferred to a substrate containing peat soil, perlite, coconut chaff, and bark (volume ratio 2:1:1:1), with a resulting survival rate of 92.2%. RAPD and ISSR markers confirmed the genetic uniformity and stability of regenerated plants.

## 1. Introduction

*Manglietiastrum sinicum* Y.W. Law, a primitive Magnoliaceae tree, is a species endemic to China, narrowly distributed in Xichou County, Yunnan province [1]. It is a popular ornamental tree known for its bright leaves, fragrant flowers, and beautiful shape; it has been described as having the appearance of the canopy of an ancient Chinese emperor, so it has the common name ‘Hua Gai Mu’ (Figure 1a) [2,3,4]. It is also valued by the timber industry for its fine-textured wood structure, which is resistant to decay and insect damage [5]. Moreover, its trunk has a compact material with a clear and delicate texture, so it is a precious and excellent tree for making furniture, wood carvings, and other crafts [6]. In addition, it is rich in essential oils [7]. However, only seven old trees remain in their natural distribution area, with no saplings or middle-aged trees [8]. This rare and important plant species is under further pressure from over-exploitation and habitat fragmentation, conditions that demand urgent steps toward conservation [4,9]. Because of these factors, this species is categorized as a Class I National Key Protected Species in China [10] and is classified as a critically endangered species (CR), according to the International Union for Conservation of Nature (IUCN) [11].

Plant tissue culture is an efficient and reliable method of clonal propagation [12]. With a lower risk of genetic instability than other regeneration methods, bud induction is considered a safe and efficient propagation method for producing plantlets from young or mature trees [13,14]. In view of these advantages, tissue culture of shoots or buds has been used by many botanic gardens in efforts to protect endangered plants [15].

Currently, some rare, high-value *Magnolia* species have been successfully propagated by in vitro culture, including *Magnolia dealbata*, *M. punduana*, and *M*. *sirindhorniae* [13,16,17,18]. However, because of the high levels of phenols found in magnolias and difficulties in rooting, the in vitro propagation of some endangered *Magnolia* plants has not been successfully carried out [19]. Moreover, various factors in the process of tissue culturing, such as medium composition and growth conditions, may lead to genetic variations among regenerated plants [20]. Thus, it is vital to ensure the genetic stability of the regenerated plants and, thereby, promote successful propagation and conservation of rare plants.

Random amplified polymorphic DNA (RAPD) and the inter-simple sequence repeats (ISSR) are polymerase chain reaction (PCR) techniques that have been successfully used in recent years to assess the genetic fidelity of regenerated plantlets in many species [20,21,22,23,24]. To date, no studies have reported either an efficient mass clonal propagation protocol for *M. sinicum*, or a genetic stability assessment of regenerated plants.

In this study, semi-lignified stem segments of *M. sinicum* were used as explants, and a protocol for axillary bud induction, bud proliferation, rooting, and acclimatization was developed and optimized. The genetic uniformity of regenerated plants was assessed by RAPD and ISSR markers. This protocol can safely be applied to large-scale propagation of this rare and valuable plant, which will make great contributions to the protection of this species.

## 2. Results

### 2.1. Explants Sterilization

After 30 days of incubation, the sterilization efficiency of soaking 1% (*v*/*v*) benzalkonium bromide sterilization solution for 8 min followed by 0.1% mercury chloride solution for 2 min was high. It effectively inhibited the propagation of endophytic bacteria (phytopathogens) and pathogens in vitro. The survival rate of explants was as high as 76%, while the contamination rate was 16%, and the mortality rate was 8%.

### 2.2. Bud Induction

After 3–7 days, most buds were initiated (Figure 1b). Different compositions and concentrations of PGRs significantly affected the induction rate and description of buds (Table 1). The optimal treatment at the bud initiation stage among the ten experiments was MS supplemented with a combination of 0.5 mg/L BA and 0.05 mg/L IBA, yielding an induction rate of up to 99.33% and vigorous green buds. The induction rate for the treatment with 0.5 mg/L BA and 0.05 mg/L NAA combination was slightly lower, with slower bud growth and fewer green buds, and the bud was easy to die after 20 days. Higher concentrations of cytokinins (BA) and auxins (IBA and NAA) led to a decrease in the induction rate; although the buds germinated quickly, they browned and died 15 days later. These results showed that IBA has a superior induction effect to NAA at the same concentration.

### 2.3. Bud Proliferation

To identify the optimal basal medium, buds initiated from explants were subcultured on seven different basal media supplemented with 0.8 mg/L BA and 0.08 mg/L IBA. The proliferation effect was significantly different in the various basal media (Table 2). We found MSM to be the optimal medium in terms of bud proliferation and elongation, inducing a 3.88-fold increase in proliferation after 30 days. The proliferation effect of DCR was the second strongest; however, the growth state of the cluster buds was poor, and the base of the plantlets tended to brown. The proliferation rate in plantlets cultured in MS, ½MS, and N6 was moderate, but bud elongation was slow. Buds of plantlets grown in WPM and B5 showed low proliferation rates and severe hyperhydricity. In contrast, the bud clusters on plantlets grown in MSM were green and healthy, without browning or hyperhydricity.

**Table 2 plants-12-02500-t002:** Effect of different basal media on bud proliferation.

Basal Medium	Proliferation Rate (%)	The Number of Buds per Explant (Length ≥ 0.5 cm)	Description
MS	257.31 ± 5.37 D	2.50 ± 0.15 C	Big crinkle leaf
MSM	388.23 ± 4.51 A	5.62 ± 0.28 A	Vigorous and green
½MS	229.73 ± 2.30 E	2.04 ± 0.11 D	Flavescent
WPM	220.10 ± 4.13 E	1.5 ± 0.16 E	Hyperhydricity
DCR	307.56 ± 13.65 B	3.82 ± 0.37 B	Browning
B5	154.24 ± 2.73 F	1.19 ± 0.10 E	Hyperhydricity
N6	274.75 ± 7.90 C	2.28 ± 0.15 D	Hyperhydricity

Notes: Different uppercase letters in the same column indicate a significant difference (*p* ≤ 0.01; Duncan’s multiple range test)In order to screen for the optimal combination and concentration of PGRs, the shoots were subcultured on MSM supplemented with different combinations of BA, KT, and IBA (Table 3). The range analysis showed that BA had the most influence on the bud number per explant. At a BA concentration of 0.4 mg/L, the proliferation rate and average number of buds were high, and the buds were vigorous and robust (Figure 1c and Figure 2a). Conversely, media with 0.6–0.8 mg/L BA induced a lower budding rate, and hyperhydric and browned buds were abundant (Figure 2c–h). Among the tested combination, the highest proliferation rate (446.20%) and average bud numbers (6.70) were obtained on MSM, containing 0.4 mg/L BA, 0.5 mg/L KT, and 0.06 mg/L IBA (Figure 1c and Figure 2a).

### 2.4. Rooting

We found a significant difference in rooting success between treatments with and without a 7-day period of darkness (Figure 3). The rooting percentage and root number were both much higher under an initial treatment with darkness than under immediate light, and the resulting roots were longer and thicker. The explants grown without an initial period of darkness and only supplemented with NAA failed to take root. However, after 7 days of culturing in darkness, the explants were able to root in MSM with NAA alone. The initial 7-day darkness treatment had positive effects on the rooting rate, as well as the state of roots.

The rooting percentage can be significantly improved by using a combination of IBA and NAA, while media supplemented with NAA alone supports a much lower rooting rate (Table 4). The rooting percentage improved significantly with an increase in IBA concentration from 0 to 1.0 mg/L. In explants treated with 7 days of darkness, adventitious roots appeared on the 15th day in MSM supplemented with 0.4 mg/L NAA and 1.0 mg/L IBA. The rooting rate was 88.89% after 30 days, and the average number of roots was 5.51 (Figure 1d,e). Moreover, the plants grown in these media were robust, with strong, long roots and green leaves. The medium with 0.4 mg/L NAA and 0.5 mg/L IBA produced the next most effective level of root induction; however, the leaves gradually turned yellow. Thus, MSM supplemented with 0.4 mg/L NAA and 1.0 mg/L IBA with a 7-day period of darkness is the optimal treatment for rooting.

### 2.5. Acclimatisation and Transplantation

The survival rate of transplanted *M. sinicum* was up to 92.2% in the substrate containing peat soil, perlite, coconut chaff, and bark at a volume ratio of 2:1:1:1. The transplanted explants under this treatment grew well after being transplanted into the substrate, the stems were robust with green and developed leaves (Figure 1f). *M. sinicum* was intolerant of waterlogging; under the meticulous management of water and fertilizer, the transplanted plants were well-adapted and grew robuantly in the nursery (Figure 1g). Since the cells of the roots were closely arranged, no aerenchyma (gas-space) formation was observed. Through transverse sections of roots of transplanted plants, we can confirm the waterlogging intolerance of *M. sinicum* (Figure 1h).

### 2.6. Assessment of Genetic Uniformity

In this study, RAPD and ISSR markers produced a total of 187 bands, with an average of 6.2 bands per locus (Table 5). The 20 RAPD primers produced a total of 116 distinct and graded bands, ranging from 200 to 2500 bp. The number of bands produced by each RAPD primer varied from 4 to 9 (Table 5). The 10 ISSR primers produced a total of 71 distinct and graded bands, ranging from 200 to 2000 bp. The number of bands produced by a single ISSR primer varied from 6 to 11. No polymorphic bands were detected between the mother plant and clonal plants as compared to negative controls, confirming the genetic fidelity of regenerated *M. sinicum* (Figure 4 and Figure 5).

## 3. Discussion

In vitro culture is affected by many factors, such as basal medium, PGRs, growth condition, etc. [25,26]. Previous studies have indicated that basal medium and PGRs are critically important factors influencing the axillary bud proliferation of magnolia species in vitro [27,28].

### 3.1. Effects of Basal Medium

The basal medium is the nutrient source for in vitro plants and plays a significant role in plant tissue culture [27]. The salt composition, especially the macronutrient component, varies throughout different basal media. Different species have different culture medium requirements [28,29,30]. One medium that is widely used for Magnoliaceae plants is MS [31]. This medium has a high salt ion concentration and abundant trace elements, with high contents of NH_4_^+^ and NO_3_^−^ [32]. However, for some woody species, the effect of MS medium is negative because of its high levels of ammonium ions [33]. Many studies have indicated that the incidence of hyperhydricity can be minimized by reducing the salt concentration of the medium, especially for endangered plants [34]. Our results showed the growth and proliferation of buds to be much higher on MSM, in which the concentrations of ammonium nitrogen (1280 mg/L) and nitrate nitrogen (2800 mg/L) were moderate. Conversely, plant performance was poor on WPM and B5, both of which contain lower concentrations of NH_4_^+^ and NO_3_^−^. Plant performance was moderate on MS and N6 media, both of which contained higher concentrations of NH_4_^+^ and NO_3,_ which would easily lead to hyperhydration [35]. The unexceptional performance of ½MS is likely due to deficiencies of K^+^, Ca^2+^, and Mg^2+^, all of which are important for meristem formation [36]. We infer that excessively high or low concentrations of NH_4_^+^ and NO_3_^−^ are not conducive to bud growth and proliferation in *M. sinicum*. It has been reported that MS with reduced salt content produces better results in many woody species than full MS [37].

### 3.2. Effects of PGRs

PGRs play a critical role in the growth of any morphogenic structure [36]. In this study, we found the synergistic effect of lower concentrations of BA (0.5 mg/L) and IBA (0.05 mg/L) to be most suitable for inducing axillary buds in bud induction. With an increase in PGR concentration, new buds were prone to browning and death in later growth periods (20th–30th day); this could be due to the high level of endogenous hormone content in *M. sinicum* [38]. Moreover, in the proliferation stage, high PGR concentration induced browning. Previous reports have indicated that a lower PGR concentration can significantly reduce phenol-induced damage in species that brown easily [39]. A low PGR concentration is sufficient to induce axillary buds in *M. sinicum.*

The ratio of cytokinin to auxin strongly influences the in vitro developmental processes. Adventitious buds were produced under high cytokinin levels in the bud proliferation stage, while adventitious roots were produced under high levels of auxin in the rooting stage [40]. The results showed that BA was essential for bud induction and was more effective than other PGRs in proliferating adventitious buds. This is likely due to the ability of BA to induce the production of natural hormones, such as zeaxanthin, which, in turn, induces organogenesis [41]. The superiority of BA for shoot proliferation has also been shown in previous studies of Magnolia species, including *Magnolia denudata* and *Liriodendron chinense* [42,43]. Many other woody plants also produce more adventitious buds when grown on media with BA than when grown with other cytokinins [44,45,46].

### 3.3. Effects of Growth Conditions

Growth conditions play a significant role in optimizing and regulating the growth of in vitro plants. Unlike PGRs, which are well-studied, the influence of light has often been overlooked. However, for species that are very difficult to propagate via tissue culture, appropriate exposure to darkness can promote the differentiation of calli, stems, buds, and roots [47]. An initial darkness treatment during rooting has been shown to be more effective than an initial treatment under light for some woody plants [48,49,50]. These previous results correspond well with the findings in this study, which demonstrate that an initial 7-day darkness treatment produces a higher rooting percentage, better growth state, and a greater number of roots.

### 3.4. Assessment of Genetic Uniformity

Stress during in vitro culture may induce mutations and cause genetic variation among regenerated plants [51]. Therefore, it is necessary to assess the genetic uniformity of the regenerated plants before confirming the success of in vitro propagation. As shown in this study, both RAPD and ISSR markers can produce their own effective polymorphic bands. The bands obtained by RAPD and ISSR primers were found to be monomorphic across all of the regenerated plants, thus confirming the genetic stability. Our results, which are consistent with previous reports, suggest that the proliferation of axillary buds minimizes the likelihood of genetic instability [23,52,53,54,55].

## 4. Materials and Methods

### 4.1. Plant Material and Explants Sterilization

Semi-lignified *M. sinicum* stem segments (5–6 cm) were collected from the South China Agricultural University (113°21′ E, 23°9′ N). After soaking in a solution of 5% (*v*/*v*) liquid detergent for 5 min, stems and leaf axils were gently scrubbed with a soft brush or cotton ball in the detergent solution. Afterward, segments were washed under running tap water for 3–4 h. They were then cut into segments (3–4 cm) with one or two buds. Surface sterilization of the explants was carried out with 1% (*v*/*v*) benzalkonium bromide sterilization solution for 8 min, followed by 0.1% mercury chloride solution for 2 min, and then rinsed with sterile distilled water six times. The number of explants was 90. After 30 days, the survival rate, contamination rate, and mortality rate of explants were observed and recorded.

### 4.2. Basal Medium and Growth Conditions

In this study, we used seven different media: (i) MS medium [56]; (ii) MSM (a special basal medium for *M. sinicum*); (iii) ½ MS; (iv) Woody Plant Medium (WPM) [57]; (v) Gamborg’s B-5 Basal Medium (B5) [58]; (vi) Douglas-fir cotyledon revised medium (DCR) [59]; and (vii) N6 medium [60]. The macronutrient component of the MSM medium is composed of the following: NH_4_NO_3_, 1280 mg/L; KNO_3_, 1520 mg/L; KH_2_PO_4_, 170 mg/L; MgSO_4_·7H_2_O, 370 mg/L; and CaCl_2_·2H_2_O, 440 mg/L. The MSM medium is modified from the standard MS medium, with the concentrations of NH_4_NO_3_ and KNO_3_ being slightly lower.

All media were adjusted to pH = 5.8, solidified with 6 g/L agar (Beijing Dingguo Changsheng Biotechnology Co., LTD., Beijing, China), and autoclaved at 121 °C for 18 min. The medium used for bud induction and proliferation contained 30 g/L sucrose, and that used for rooting contained 15 g/L sucrose [61]. The cultures were incubated in the laboratory at 25 ± 2 °C with a relative humidity of 60–70% and a 12 h/d illumination cycle under a cool white light provided by fluorescent lamps (1500–2000 Lx).

### 4.3. Bud Induction

After sterilization, the stems with buds were cut and transferred into induction media. To compare the relative induction effects of different PGRs, we chose MS as the basal medium for bud induction and applied various concentrations of different PGRs: 6-benzylaminopurine (BA) 0, 0.5, 1.0, 1.5, 2.0 mg/L; indole-3-butyric acid (IBA) 0, 0.05, 0.1, 0.15, 0.2 mg/L; or 1-naphthylacetic acid (NAA) 0, 0.05, 0.1, 0.15, 0.2 mg/L (Table 1). In total, ten treatments were carried out. Each treatment had three replications with a total of 90 explants. After 20 days, the percentage of bud induction, time taken for bud initiation (marked by the separation layer on the edges of the petiole), and the growth state were recorded. The culture container for bud induction is a 100–mL conical flask.

### 4.4. Bud Proliferation

Nodal segments (1–2 cm) were removed and transferred into seven different basal culture media with the same PGRs (0.8 mg/L BA and 0.08 mg/L IBA), the concentration of BA and IBA were chosen according to the pre-experiments of prolifer-ation (Appendix A); these were compared during the sub-culture phase (Table 2).

In order to optimize the rate of bud proliferation, MSM was supplemented with different concentrations of BA (0.1, 0.3, and 0.5 mg/L) in combination with kinetin (KT) (0.05, 0.1, and 0.3 mg/L) and IBA (0.01, 0.03, and 0.05 mg/L; Table 3). In total, nine treatments were carried out. Each treatment had three replications with a total of 90 explants. After 30 days, the proliferation rate and number of new buds per explant (≥0.5 cm) were recorded. The culture container for bud proliferation and rooting is a 240–mL tissue culture flask.

### 4.5. Rooting

Robust shoots (≥1.2 cm in height) were removed and transferred to the rooting medium. To optimize root induction, MSM was supplemented with different compositions and concentrations of the PGRs: NAA (0.4–1.2 mg/L) and IBA (0–1.0 mg/L) (Table 4). Two groups were set, with the same PGR treatment but different light treatments. The first group was transferred to light culture after dark treatment for 7 days, while the second group was cultured immediately under light. In total, 18 treatments were carried out. Each treatment had three replications with a total of 90 explants. The percentage of root induction, root numbers, and the growth state of roots were observed and recorded after 30 days.

### 4.6. Acclimatisation and Transplantation

After 30 days in rooting culture, the plantlets with well-developed roots were placed in the greenhouse for 7 days. The temperature of the greenhouse was 25 ± 1 °C with relative humidity of 70–80%. The lid of each culture bottle was removed 2 days before transplanting in order to allow the plantlets to adjust to the greenhouse environment. Plantlets were gently removed from the culture vessels, and any adhering medium and loose callus were washed off. Plantlets were then transplanted into a substrate containing peat soil, perlite, coconut chaff, and bark at a volume ratio of 2:1:1:1, which had been previously disinfected with potassium permanganate solution [1000–1250 ppm]. Plantlets were watered thoroughly and then covered with transparent plastic film and a 70% shade cloth. The humidity and air permeability were replenished daily; rotten seedlings and leaves were removed, and insect pests were controlled. The film and shade cloth were removed after 10 days when we conducted routine water and fertilizer management. The number of plantlets was 90. Survival rates and growth conditions were recorded 60 days after transplantation.

### 4.7. Assessment of Genetic Uniformity

To assess genetic fidelity, total genomic DNA was extracted from fresh leaves (including 18 randomly selected clones and their parent plants) using the cetyltrimethylammonium bromide (CTAB) method [62]. The total genomic DNA of a non-clonal *M. sinicum* seedling was also extracted as a negative control. The concentration of DNA was measured using a NanoDrop 2000 spectrophotometer (Thermo Fischer Scientific, Waltham, MA, USA). For RAPD analysis, a total of 20 primers (TsingKe Biological Technology, Tianjin, China) were used according to previous reports and initial experiments [63,64]. The ISSR analysis was performed with ten ISSR primers (TsingKe), which were selected for genetic analysis of magnolia plants based on previous reports [65,66].

DNA amplification for RAPD and ISSR markers was performed in a volume of 10 μL reaction mixture containing 0.5 μL of template DNA (20 ng), 5 μL of 2× Taq Plus MasterMix (Beijing ComWin Biotech Co., Ltd., Beijing, China), 0.5 μL primer, and 4 μL ddH_2_O. RAPD was performed using a thermal cycler (Bio-Rad, Hercules, CA, USA) programmed for initial denaturation at 94 °C for 5 min, followed by 40 cycles of denaturation at 94 °C for 45 s, annealing at 40 °C for 45 s, and extension at 72 °C for 90 s, with a final extension at 72 °C for 10 min. ISSR amplification was performed using a thermal cycler (Bio-Rad), programmed for initial denaturation at 94 °C for 3 min, followed by 35 cycles of denaturation at 94 °C for 30 s, annealing at 50 °C for 30 s, and extension at 72 °C for 30 s with a final extension at 72 °C for 3 min. All PCR procedures were repeated three times under the same conditions in order to verify the accuracy of the amplified products. Amplified products were electrophoresed in 1.5% agarose gel containing 0.25 μg/mL EB (Invitrogen, Carlsbad, CA, USA) using 1× TAE (Tris Acetate EDTA) buffer. The size of the amplification products was estimated with a 100 bp DNA ladder or a 5000 bp DNA marker (Takara, Kyoto, Japan). The gels were photographed using the gel documentation system (Bio-Rad, Hercules, CA, USA); only clear DNA bands were considered.

### 4.8. Statistical Analysis

Induction rate (%) = the number of induced explants/the number of total initial explants × 100%. Proliferation rate (%) = the total number of buds (length ≥ 0.3 cm)/the number of initial buds on the sub-cultured explants × 100%. Effective shoot rate (%) = the total number of buds (length ≥ 0.5 cm)/the number of initial buds on the sub-cultured explants × 100%. Rooting rate (%) = the number of rooted plantlets/the number of total shoots × 100%. Root numbers = the total number of roots/the number of rooted seedlings. The software SPSS v23.0 [67] was used for the statistical analyses. Significant differences among means were compared using Duncan’s multiple range test at *p* ≤ 0.05 and *p* ≤ 0.01; the results were represented as mean ± standard error of three replicates.

## 5. Conclusions

This is the first successful in vitro propagation of *M. sinicum* from mature tree stems on a large scale without browning. Direct multiple bud induction from axillary buds reduces the risk of genetic instability. We found that MS supplemented with 0.5 mg/L BA and 0.05 mg/L IBA is the optimal medium for bud initiation, with an induction percentage of 99.33%. The maximum bud proliferation rate (446.20%) was obtained on MSM fortified with 0.4 mg/L BA, 0.5 mg/L KT, and 0.06 mg/L IBA. The MSM supplemented with 0.4 mg/L NAA and 1.0 mg/L IBA was shown to be the best for rooting, with an average of 5.51 roots for each plant. After an initial 7-day darkness treatment, in vitro plants started to root after 15 days, with the rooting rate reaching 88.89% by the 30th day. The regenerated plantlets acclimatized well to a natural setting, with a survival rate of 92.2%. RAPD and ISSR markers confirmed the genetic uniformity of regenerated plants. Hence, this protocol can be used for the efficient propagation and conservation of *M. sinicum*.

## Figures and Tables

**Figure 1 plants-12-02500-f001:**
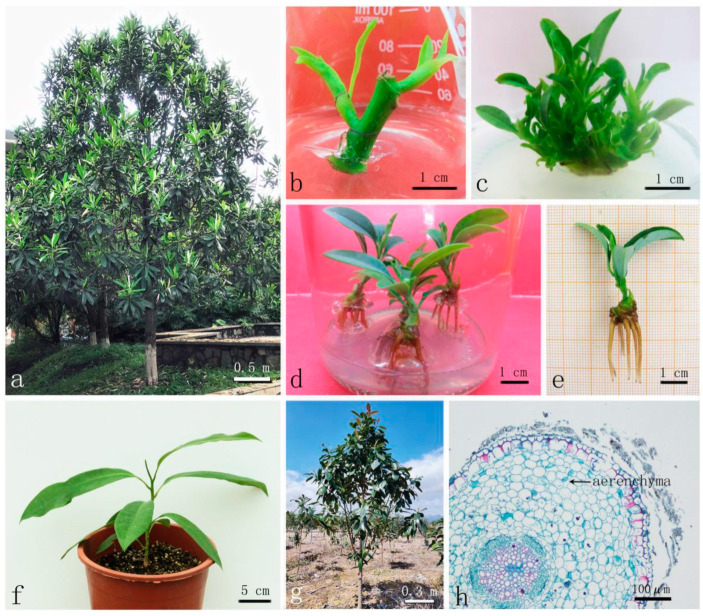
In vitro propagation of *Manglietiastrum sinicum* using mature axillary node explants. (**a**) Mature tree. (**b**) Bud initiation for 20 days (in the medium MS + 0.5 mg/L BA + 0.05 mg/L IBA). (**c**) Bud proliferation for 30 days (in the medium MSM + 0.4 mg/L BA + 0.5 mg/L KT + 0.06 mg/L IBA). (**d**,**e**) Rooting for 30 days (in the medium MSM + 0.4 mg/L NAA + 1.0 mg/L IBA, with initial 7-day darkness treatment). (**f**) Acclimatized plants grow well after being transplanted into substrate for 60 days (containing peat soil, perlite, coconut chaff, and bark with a volume ratio of 2:1:1:1). (**g**) Transplanted plants are well-adapted. (**h**) Transverse sections of roots of transplanted plants.

**Figure 2 plants-12-02500-f002:**
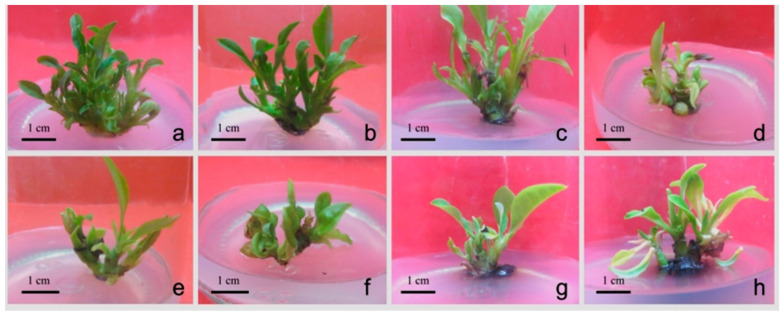
The growth state of bud under different treatments. (**a**) Green, vigorous, and robust buds. (**b**) Green buds. (**c**,**d**) Yellow buds. (**e**,**f**) Hyperhydricity buds. (**g**) Browning buds; (**h**) Yellow and browning buds.

**Figure 3 plants-12-02500-f003:**
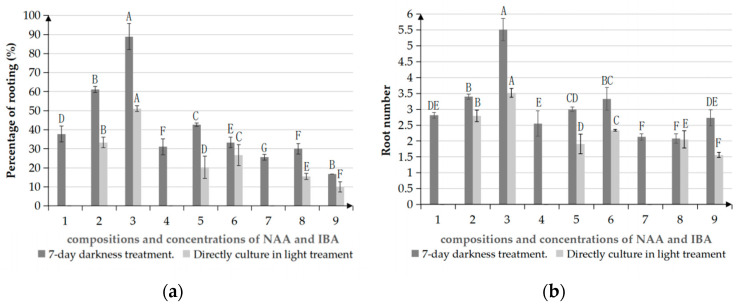
Effect of initial darkness treatment on the rooting of *Manglietiastrum sinicum* under different auxin treatments. Group One was transferred to light culture after a 7-day darkness treatment. Group Two was cultured in light for the entire 30-day period. (**a**) Percentage of rooting. (**b**) Root number per explant. Different uppercase letters in the same column indicate a significant difference (*p* ≤ 0.01; Duncan’s multiple range test).

**Figure 4 plants-12-02500-f004:**
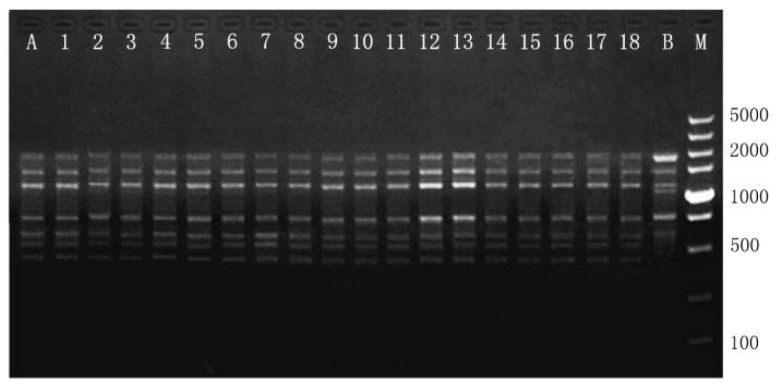
RAPD profiles generated by PCR amplification with primer S167. Lane M: Molecular marker (100–5000 bp for S167). Lane A: A non-clonal *Manglietiastrum sinicum* plant developed from seed (negative control). Lanes 1–18: Regenerated plants. Lane B: Mother plant.

**Figure 5 plants-12-02500-f005:**
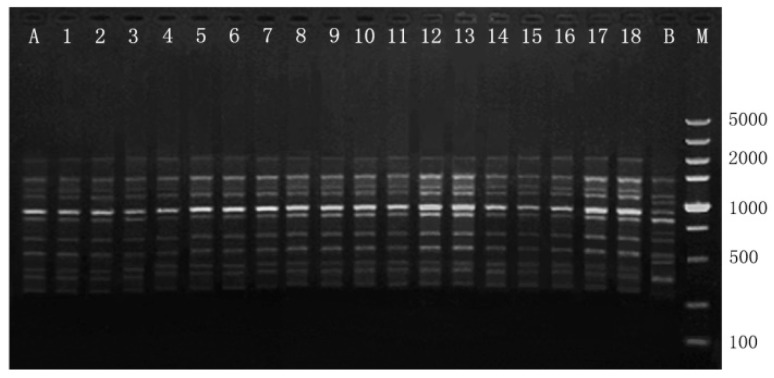
ISSR profiles generated by PCR amplification with primer UBC811. Lane M: Molecular marker (100–5000 bp for UBC811). Lane A: A non-clonal *Manglietiastrum sinicum* plant developed from seed (negative control). Lane 1–18: Regenerated plants. Lane B: Mother plant.

**Table 1 plants-12-02500-t001:** Effect of different compositions and concentrations of plant growth regulators on bud induction.

BA mg/L	IBA mg/L	NAA mg/L	Induction Rate (%)	Initiation Time	Description
0	0	0	75.57 ± 2.84 E	10th day	Green buds
0.5	0.05	0	99.33 ± 0.47A	6th day	Robust and green buds
1.0	0.1		91.11 ± 3.14 B	5th day	Yellow buds
1.5	0.15		82.22 ± 4.16 D	3rd day	Hyperhydricity and browning
2.0	0.2		65.17 ± 1.38 F	3rd day	Hyperhydricity and browning
0.5	0	0.05	97.74 ± 1.60 A	7th day	Yellow buds
1.0		0.1	86.55 ± 2.58 C	5th day	Green buds
1.5		0.15	75.56 ± 3.14 E	3rd day	Hyperhydricity buds
1.5		0.15	74.0 ± 0.82 E	3rd day	Hyperhydricity and browning
2.0		0.2	52.7 ± 1.25 G	3rd day	Hyperhydricity and browning

Notes: Different uppercase letters in the same column indicate a significant difference (*p* ≤ 0.01; Duncan’s multiple range test).

**Table 3 plants-12-02500-t003:** Effect of different compositions and concentrations of BA, KT, and IBA on bud proliferation and range analysis.

BA mg/L	KT mg/L	IBA mg/L	Proliferation Rate (%)	The Number of Buds per Explant (Length ≥ 0.5 cm)	Description
0	0	0	112.25 ± 4.15 G	0.76 ± 0.22 H	Browning
0.4	0.3	0.04	322.41 ± 3.97 B	4.14 ± 0.12 B	Vigorous and green
0.4	0.5	0.06	446.20 ± 9.90 A	6.70 ± 0.51 A	Vigorous and robust
0.4	0.8	0.08	265.30 ± 5.79 D	3.52 ± 0.14 C	Green buds
0.6	0.3	0.06	293.50 ± 3.30 C	3.92 ± 0.51 B	Green buds
0.6	0.5	0.08	271.23 ± 10.11 D	3.16 ± 0.19 D	Browning
0.6	0. 8	0.04	219.37 ± 3.68 E	1.86 ± 0.03 F	Browning
0.8	0.3	0.08	284.22 ± 3.91 C	2.54 ± 2.84 E	Browning
0.8	0.5	0.06	206.93 ± 7.60 E	1.58 ± 2.07 F	Hyperhydricity
0.8	0.8	0.04	179.77 ± 5.36 F	1.23 ± 0.29 G	Browning

Notes: Different uppercase letters in the same column indicate a significant difference (*p* ≤ 0.01, Duncan’s multiple range test).

**Table 4 plants-12-02500-t004:** Effects of auxin on rooting with 7-day darkness treatment.

No.	NAA mg/L	IBA mg/L	Percentage of Rooting (%)	Root Number	Description
1	0.4	0	37.78 ± 4.16 D	2.81 ± 0.09 DE	Short roots, short plantlets
2	0.4	0.5	61.11 ± 1.57 B	3.40 ± 0.08 B	Short roots, flavescent leaf
3	0.4	1	88.89 ± 6.85 A	5.51 ± 0.35 A	Strong and long roots, vigorous and green plantlets
4	0.8	0	31.11 ± 4.16 F	2.55 ± 0.40 E	Short plantlets
5	0.8	0.5	42.68 ± 0.92 C	3.00 ± 0.07 CD	Short and thin roots
6	0.8	1	33.33 ± 2.72 E	3.33 ± 0.36 BC	Short and thin roots, short plantlets
7	1.2	0	25.56 ± 1.57 G	2.13 ± 0.10 F	Plenty of callus, flavescent leaf
8	1.2	0.5	30.00 ± 2.72 F	2.08 ± 0.15 F	Short roots
9	1.2	1	16.67 ± 0.00 H	2.73 ± 0.25 DE	Strong and long roots

Notes: Different uppercase letters in the same column indicate a significant difference (*p* ≤ 0.01; Duncan’s multiple range test).

**Table 5 plants-12-02500-t005:** List of primers, their sequences, number of bands, and size of the amplified fragments generated by 20 RAPD and 10 ISSR primers.

Primers	Primer Sequence (5′–3′)	No. of Bands	Range of Amplification (bp)
RAPD			
S10	CTGCTGGGAC	5	200–2000
S11	GTAGACCCGT	6	300–1500
S18	CCACAGCAGT	6	300–1200
S22	TGCCGAGCTG	4	200–1000
S24	AATCGGGCTG	6	300–1200
S30	GTGATCGCAG	3	600–1000
S31	CAATCGCCGT	9	400–1500
S38	AGGTGACCGT	7	400–1200
S40	GTTGCGATCC	4	500–1500
S69	CTCACCGTCC	5	300–1000
S144	GTGACATGCC	8	150–1200
S154	TGCGGCTGAG	6	500–2500
S155	ACGCACAACC	6	500–2000
S158	GGACTGCAGA	4	400–1500
S160	AACGGTGACC	7	200–1200
S163	CAGAAGCCCA	8	200–1500
S167	CAGCGACAAG	7	400–2000
S17	AGGGAACGAG	7	400–1500
S173	CTGGGGCTGA	4	500–1000
S174	TGACGGCGGT	4	400–1000
Total		116	
ISSR			
UBC811	GAGAGAGAGAGAGAGAC	9	300–1500
UBC835	AGAGAGAGAGAGAGAGYC	11	300–2000
UBC840	GAGAGAGAGAGAGAGACTT	8	350–1200
UBC842	GAGAGAGAGAGAGAGACTG	8	400–1500
UBC844	CTCTCTCTCTCTCTCTRC	5	500–2000
UBC847	CACACACACACACACARC	7	400–2000
UBC848	CACACACACACACACARG	5	300–1500
UBC855	ACACACACACACACACYT	6	200–1500
UBC857	ACACACACACACACACYG	7	400–2000
UBC864	ATGATGATGATGATGATG	5	500–1200
Total		71	

## Data Availability

The data supporting the report results can be found in the attachment.

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
