# Peer review of "In Vitro Propagation and Genetic Uniformity Assessment of Manglietiastrum sinicum: A Critically Endangered Magnoliaceae Species"

_plants, 2023, doi:10.3390/plants12132500_

Round 1

Reviewer 1 Report

L23 MSM medium (a special basal medium for M………………………………………….

L92 In this study, we used seven different media: (i) MS medium [21], (ii) MMS (a special……….

In the two sections above, is there a difference between MSM and MMS?

Author Response

Point 1: L23 MSM medium (a special basal medium for M…
L92 In this study, we used seven different media: (i) MS medium [21], (ii) MMS (a special…In the two sections above. is there a difference between MSM and MMS?
Response 1: There is no difference between MSM and MMS medium. Actually the name of the basal medium is MSM, I'm so sorry for the mistake. Has been modified in L92.

Reviewer 2 Report

Dear Authors,

I find your manuscript valuable for publication and generally properly written. Introduction provides sufficient background and aim pointing, methods are properly appllied and described, results are clearly presented, discussion and conclusions are concise and well interpret.

My only concerns are:

line 38 appearance....of an ancient emperor .... not clear for me, precise please

55-latin plant names in italics

M&M 2.1 provide nformation on the culture vessel

101 - why did you lower the succros concentration for rooting? Can you refer to any experiments confirming the lowering succrose conc. is beneficial for rooting?

103 -should be: "cool white light provided by flourescent lamps"

Results - give a verse about the sterilization efficiency

check tables formatting in the journal, did you use template?

Fig.2 should be Directly cultured in light

Regards

Generally good English quality.

Author Response

Point 1: L 38 appearance....of an ancient emperor .... not clear for me, precise please

Response 1: Seen from a distance, Manglietiastrum sinicum shapes like the canopy of an ancient Chinese emperor, so it has the Chinese name Hua Gai Mu. “Hua” means ancient Chinese emperor, “Gai” means canopy, “Mu” means tree. So it has been described as having the appearance of the canopy of an ancient Chinese emperor. Modified in L38-39.

Point 2: L55-latin plant names in italics

Response 2: Has been modified in L55-56. I'm so sorry for the mistake.

Point 3: M&M 2.1 provide information on the culture vessel.

Response 3: I’m not sure what this question means. May I ask what kind of information should I provided?

Point 4: L101 - why did you lower the succros concentration for rooting? Can you refer to any experimentsconfirming the lowering succrose conc. is beneficial for rooting?

Response 4: The concentration of sucrose was reduced during the rooting process in plant tissue culture for several reasons:

  1. Adventitious roots were not induced in the medium with 3% sucrose concentration in the preliminary experiment.
  2. Promoting root development: High concentrations of sucrose maybeinhibit or delay root initiation and development. By reducing the sucrose concentration, it encourages the plant tissue to focus its energy on root growth.
  3. Minimizing osmotic stress:By reducing the sucrose concentration, osmotic stress is minimized, allowing for better root growth.

Here are references that demonstrate experiments involving the reduction of sucrose concentration during adventitious root induction:

  1. A study titled "Not all sugars are sweet for banana multiplication. In vitro multiplication, rooting, and acclimatization of banana as influenced by carbon source-concentration interactions" by Waman, A.A. et al., published in the journal In Vitro Cell.Dev.Biol.-Plant (2014), investigated the effect of different sucrose concentrations on adventitious root formation in banana shoots. The researchers found that sucrose levels affected root induction and subsequent development. https://doi.org/10.1007/s11627-014-9623-3
  2. In a study published in the Journal of Chinese Horticultural Abstracts (2015) by Lv et al., titled "Study on Adventitious Root Induction of Tissue Culture Seedlings of Alpinia zerumbet" ,the authors explored the impact of sucrose concentration on adventitious root formation in Alpinia zerumbet. They observed that lowering sucrose levels to 1-2% significantly enhanced root induction and subsequent development.

Point 5: 103 -should be: "cool white light provided by flourescent lamps"

Response 5: Has been modified.

Point 6: Results - give a verse about the sterilization efficiency

Response 6: Has been added in Results -3.1.explants sterilisation.

Point 7: check tables formatting in the joumal, did you use template?

Response 7: Has been checked and modified.

Point 8: Fig.2 should be Directly cultured in light.

Response 8: Fig. 2 has been modified.

Reviewer 3 Report

Presented article is designed according to the rules of the journal, contains all the necessary sections. The article is devoted to the development of the protocol of microclonal multiplication for the conservation of valuable endemic species Manglietiastrum sinicum Y.W. Law. The stated goal and the tasks set by the authors of the article have been fulfilled by the authors and are reflected in the article.

The authors should correct only a few minor remarks:

 Add to the abstract the result on the shade treatment of plants In vitro;

Move Table 3 after the reference to this table in the text (line 218).  Also in the title of Table 3 indicate the medium in which BA, KT and IBA compounds were added;

Increase the scale of Fig. 2, in the present version the figure is poorly readable;

Explain why fragment g "Transverse sections of roots of transplanted plants" is shown in Fig. 1? Or remove it from figure 1.

Perhaps table 5 should be sent to Supplements.

The presented article can be published in a journal.

Author Response

Point 1: Add to the abstract the result on the shade treatment of plants In vitro.

Response 1: Has been added in abstract (L27-28).

Point 2: Move Table 3 after the reference to this table in the text (line 218). Also in the title of Table 3 indicate the medium in which BA KT and IBA compounds were added.

Response 2: Table 3 has been moved. But the title of Table 3 has been indicate the BA KT and IBA compounds , I’m not sure what “Also in the title of Table ndicate the medium in which BA KT and IBA compounds were added.”mean.

Point 3: increase the scale of Fig. 2, in the present version the figure is poorly readable;.

Response 3: Fig. 2 has been modified.

Point 4: Explain why fragment g "Transverse sections of roots of transplanted plants" is shown in Fig. 1?Or remove it from figure 1.

Response 4: Manglietiastrum sinicum was intolerant of waterlogging, under the meticulous management of water and fertilizer, the transplanted plants were well adapted and grew robuantly in the nursery (Figure 1g). Through transverse sections of roots of transplanted plants, we can confirm the waterlogging intolerance of M. sinicum (Figure 1h). Since the cells of roots were closely arranged, and no aerenchyma(Gas-space) formation was observed. Therefore, it has strong drought resistance and waterlogging intolerance. In the transplanting management, it is necessary to choose the substrate with good water permeability and pay attention to the control of soil moisture. This is an important information for the successful of transplanting.

It have been added in the manuscript in L267-L273.

Point 5: Perhaps table 5 should be sent to Supplements

Response 5: I think this is an important information in the article, supporting the assessment of genetic uniformity in the manuscript, such as different kind of primers, corresponding number of  bands and range of amplification.
